# A Novel Mitochondrial Genome Fragmentation Pattern in the Buffalo Louse *Haematopinus tuberculatus* (Psocodea: Haematopinidae)

**DOI:** 10.3390/ijms232113092

**Published:** 2022-10-28

**Authors:** Yi-Tian Fu, Chaoqun Yao, Hui-Mei Wang, Wei Wang, Guo-Hua Liu

**Affiliations:** 1Research Center for Parasites & Vectors, College of Veterinary Medicine, Hunan Agricultural University, Changsha 410128, China; 2Department of Zoology, University of Swabi, Swabi 23561, Pakistan; 3Department of Biomedical Sciences and One Health Center for Zoonoses and Tropical Veterinary Medicine, Ross University School of Veterinary Medicine, Basseterre P.O. Box 334, Saint Kitts and Nevis; 4The Centre for Bioinnovation, School of Science and Engineering, University of the Sunshine Coast, Sippy Downs, QLD 4556, Australia

**Keywords:** buffalo louse, fragmented mt genome, recombination, phylogenetic analyses

## Abstract

Sucking lice are obligate ectoparasites of mammalian hosts, causing serious public health problems and economic losses worldwide. It is well known that sucking lice have fragmented mitochondrial (mt) genomes, but many remain undetermined. To better understand patterns of mt genome fragmentation in the sucking lice, we sequenced the mt genome of the buffalo louse *Haematopinus tuberculatus* using next-generation sequencing (NGS). The mt genome of *H. tuberculatus* has ten circular minichromosomes containing a total of 37 genes. Each minichromosome is 2.9–5.0 kb long and carries one to eight genes plus one large non-coding region. The number of mt minichromosomes of *H. tuberculatus* (ten) is different from those of congeneric species (horse louse *H. asini*, domestic pig louse *H. suis* and wild pig louse *H. apri*) and other sucking lice. Two events (gene translocation and merger of mt minichromosome) are observed in *Haematopinus*. Compared to other studies, our phylogeny generated from mt genome datasets showed a different topology, suggesting that inclusion of data other than mt genomes would be required to resolve phylogeny of sucking lice. To our knowledge, this is the first report of a ten mt minichromosomes genome in sucking lice, which opens a new outlook into unexplored mt genome fragmentation patterns in sucking lice.

## 1. Introduction

The sucking lice (Psocodea: Anoplura) are obligate ectoparasites of eutherian mammals. There are approximately 540 known species in 15 families [1]. The single genus family Haematopinidae contains 21 described species, which are important ectoparasites of domestic animals that cause significant economic losses [2,3]. In addition, *Haematopinus* species are vectors of several pathogens, such as African swine fever virus [4], swinepox virus [5], classical swine fever virus [6] and *Anaplasma* spp. [7]. 

Metazoan mitochondrial (mt) genomes are usually circular DNA molecules of 13–20 kb with 36–37 genes including 12–13 protein-coding genes, two rRNA genes and 22 tRNA genes [8,9,10]. However, mt genomes of eutherian mammalian lice and some avian lice exhibit diverse fragmentation patterns. An example of an extremely fragmented mt genome is the human body louse *Pediculus humanus humanus* with 20 mt minichromosomes [11]. To date, the mt genomes of 21 sucking lice species (12 complete mt genomes and 9 incomplete mt genomes) have been sequenced, all are extensively fragmented with different numbers of minichromosomes [11,12,13,14,15,16,17,18,19,20,21,22]. Often, mt gene arrangement and composition are stable among members of a louse genus [12,17,19]; however, substantial variation in mt karyotype, the number of mt minichromosomes, gene arrangement and gene content has been also reported among congeneric lice, including sucking lice. In primate lice, the human louse *P. humanus* and chimpanzee louse *P. schaeffi* have 20 and 18 minichromosomes, respectively [12,17]. Macaque louse *Pedicinus obtusus* and colobus louse *P. badii* have 12 and 14 minichromosomes, respectively [19]. The goat louse *Bovicola caprae* of 13 minichromosomes is different from cattle louse *B. bovis* and sheep louse *B. ovis* (12 minichromosomes) [23]. Furthermore, variation in congeneric avian lice has also been found in pigeon lice, *Columbicola columbae*, *C. macrourae*, *C. passerinae* 1 and *C. passerinae* 2 have 15, 16, 17 and 17 minichromosomes, respectively [24]. Conversely while the horse louse *H. asini*, pig louse *H. suis* and wild pig louse *H. apri* all have nine mt minichromosomes [13,16], gene content and gene order of three minichromosomes in *H. asini* differ from those of *H. suis* and *H. apri*. Interestingly, the mt genomes of both *H. suis* and *H. apri* also have tRNA pseudogenes [13]. Based on these findings, we hypothesize that various mt genome fragmentation patterns exist in the genus *Haematopinus*. However, this hypothesis is built on only three *Haematopinus* species [13,16], thus, there is a need to obtain more mt genomes to test this hypothesis. 

To further explore mt genome evolution in *Haematopinus*, we used next-generation sequencing (NGS) on *H. tuberculatus*. We found that the mt genome of *H. tuberculatus* is fragmented into ten circular minichromosomes. We analyzed mt genome fragmentation pattern and phylogeny, as well as variation in mt minichromosome composition and recombination with in the genus *Haematopinus*. Our results are invaluable in understanding the evolution of fragmented mt genomes in the sucking lice.

## 2. Results and Discussion

### 2.1. General Features of the mt Genome of the Buffalo Louse H. tuberculatus

Sequencing the *H. tuberculatus* genome produced 3.4 Gb of Illumina short-read sequence data, a total of 6,710,412 × 2 raw reads. After quality filtration, 3,841,215 × 2 clean reads were suitable for assembly of the mt genome. Assembling these sequence-reads into contigs, identified all 37 mt genes including 13 protein-coding genes, 22 tRNA genes and two rRNA genes, typical of bilateral animals. There are ten minichromosomes (Figure 1; Table 1); each minichromosome is 2.9–5.0 kb in size and consists of a coding region and one non-coding regions (NCR) (Table 1). The coding regions have 1–8 genes each and vary in size from 67 bp to 2627 bp (Table 1). All genes are in identical orientation relative to the transcription origin except *trn*T, *nad*1 and *trn*Q (Figure 1). The raw data (BioProject accession number: PRJNA883441) and nucleotide sequences (GenBank accession numbers: ON416547-56) of *H. tuberculatus* have been deposited in the NCBI database.

The NCR is composed largely of motifs conserved between different minichromosomes [24], and this region includes the D-loop which is involved in DNA replication and the initiation of transcription [25]. We assembled the full-length NCRs for all mt minichromosomes of *H. tuberculatus*, which ranged from 2280 bp (*trn*H-*nad*5-*trn*F-*nad*6 minichromosome) to 2901 bp (*trn*R-*nad*4L minichromosome) in size (Table 1). The longest NCR (2901 bp) in the buffalo louse *H. tuberculatus* is shorter than that in the horse louse *H. anisi* (3264 bp) [16], while it is longer than those of other sucking lice. As in most parvorder Anoplura, there is a GC-rich motif (70 bp, 55.7% C and G) downstream of the 3′-end of the coding region in each NCR. Remarkably, the AT-rich motif (54 bp, 94.4% A and T) is in the middle of the NCR, rather than upstream of the 5′-end of coding region, differing from other parvorder Anoplura [11,12,13].

### 2.2. Numbers of Minichromosomes among Parvorder Anoplura

All sucking lice sequenced to date have fragmented mt genome with variable numbers of, i.e., 9, 11, 12, 14, 18 or 20 minichromosomes. All mt genes have been identified in each of the 13 complete mt genomes, each circular minichromosome comprises one coding region and one NCR (Figure 2). An additional nine incomplete mt genomes of sucking lice are shown in Appendix A. Previous studies have suggested that fragmented mt minichromosomes are under strong selection to remain functional, and the related function may be affected along with an increased number of mt minichromosomes [26,27]. In the present study, we identified a novel pattern in the mt genome of *H. tuberculatus* with ten minichromosomes. Previous studies have indicated that the number of mt minichromosomes is evolutionarily unstable across Anoplura, even between congeneric species [12,17,19]. The substantial variation in the number of mt minichromosomes among Anoplura suggests that the process of mt genome fragmentation is a continuous process. 

### 2.3. Variation in mt Minichromosomal Composition among Haematopinus Lice

*H. suis*, *H. apri* and *H. asini* each have nine mt minichromosomes [13,16]; however, *H. tuberculatus* has ten. The distribution of genes across the nine minichromosomes is identical between *H. suis* and *H. apri* [13]. Six minichromosomes in *H. tuberculatus* have the same gene content and gene order as their counterparts in *H. asini*, *H. suis* and *H. apri*, but the remaining differs [13,16]. In *H. suis/apri*, one minichromosome carries four genes, i.e., *trn*R-*nad*4L-*nad*6-*trn*M, which are found on three separated minichromosomes in *H. tuberculatus* (Figure 3) and *H. asini* [16]. In *H. tuberculatus* and *H. asini*, one minichromosome has four genes, *trn*H-*nad*5-*trn*F-*nad*6 (Figure 3), however, in *H. suis* and *H. apri*, the corresponding minichromosome has only three genes, *trn*H-*nad*5-*trn*F [13]. In *H. tuberculatus*, *H. suis* and *H. apri*, one minichromosome has two genes, *rrn*S-*trn*C (Figure 3). In contrast, this minichromosome in *H. asini* has four genes, *trn*R-*nad*4L-*rrn*S*-trn*C. In *H. tuberculatus* and *H. asini*, *trn*M occurs on its own minichromosome (Figure 3), however, in *H. suis* and *H. apri*, *trn*M is along with *trn*R-*nad*4L and *nad*6. These results clearly show the substantial variation in mt karyotype among *Haematopinus* species. Several previous studies compared mt genomes between the lice within the same genus, and showed substantial variation in mt karyotypes [14,19,20,21,22,23,24]. Taken together, these studies indicate that intra-genus variation in mt minichromosome composition is common in lice.

### 2.4. Recombination of mt Minichromosomes in the Haematopinus Lice

Recombination has been proposed as a possible mechanism contributing to the evolution of mt genome fragmentation across animal clades [27]. Long identical nucleotide sequences ranging from 14 to 133 bp are shared between mt genes, providing evidence for recombination between mt minichromosomes in sucking lice [11,12,13,14,15,16,17]. Similarly, seven stretches of identical nucleotide sequences, 7 to 32 bp long, were found between five pairs of mt genes in the buffalo louse (Table 2). *trn*L_1_ and *trn*L_2_ share three stretches of identical sequences of 7, 11 and 25 bp long. Meanwhile, in pig lice, the two genes share 9, 10 and 16 bp long identical sequences, whereas in horse louse these two genes share 15 bp long identical sequences with one another (Table 2). *rrn*L and *rrn*S share two stretches of identical sequences, 10 and 33 bp long, in *H. asini*; however, in *H. suis, H. apri* and *H. tuberculatus*, these two genes share only one stretch, 9 to 11 bp long of identical sequence, suggesting that recombination is occasional (Table 2). Previous studies found that recombination among tRNA genes could affect tRNA secondary structures [12,16]. Among *Haematopinus*, in addition to the pair of tRNA genes mentioned above, *trn*P and *trn*T share longer identical sequences than expected in *H. suis* (26 bp), *H. apri* (26 bp) and *H. asini* (27 bp), respectively. Nevertheless, in *H. tuberculatus*, the two genes only share 7 bp long identical sequences as other sucking lice do (Table 2). Among protein-coding genes, *atp*8 and *atp*6 in *H. tuberculatus* share a 32 bp identical sequence, three to four times more than expected by chance (Table 2), *nad*4 and *cox*1 share a 19 bp identical sequence in *H. tuberculatus*, and share 12 and 18 bp identical sequences in *H. suis*, but do not share longer-than expected identical sequences in *H. asini* and *H. apri*. Meanwhile, *nad*4 and *cyt*b share 17 bp in *H. tuberculatus*; and 20 bp in *H. asini*, but no longer-than expected identical sequences are seen in *H. suis* and *H. apri*, nor in other sucking lice (Table 2). There is a 14 bp identical sequence shared by *nad*4L and *trn*V genes in *H. tuberculatus*, which is approximately twice as in other sucking lice. These results indicate that recombination is a likely cause of shared identical sequences between mt genes in *Haematopinus* lice. 

Gene translocation between mt minichromosomes has been reported in the horse louse *H. asini* [16] and in the shrew louse *P. reclinate* [21], indicating that it is common in sucking lice. In the present study, translocations in *Haematopinus* lice can be also accounted for by two events of recombination. Firstly, in *H. suis/apri*, a *nad*6 moved from the *trn*R-*nad*4L-*nad*6-*trn*M minichromosome to the *trn*H-*nad*5-*trn*F to generate a new minichromosome *trn*H-*nad*5-*trn*F-*nad*6 in *H. tuberculatus* (Figure 1) and *H. asini* [16]. Second, in *H. suis/apri*, *trn*R-*nad*4L transferred from the minichromosome that contained *trn*R-*nad*4L-*nad*6-*trn*M, while in *H. tuberculatus*, *trn*R-*nad*4L moved to *rrn*S-*trn*C to generate a minichromosome, *trn*R-*nad*4L-*rrn*S-*trn*C in *H. asini* (Figure 1). These results suggest that recombination resulted in gene translocation between mt minichromosomes.

Previous studies have showed that merging and splitting occur between the minichromosomes of sucking lice [18,19,22]. Specifically, mergers but no split have been previously observed in *Haematopinus* spp. Sequenced [18]. Two mergers occurred in *H. tuberculatus* in the current study. First, the ancestral minichromosomes, *trn*K-*nad*4 and *atp*8-*atp*6-*trn*N, merged into one minichromosome, *trn*K-*nad*4-*atp*8-*atp*6-*trn*N (Figure 4). Second, *nad*2 and *trn*I-*cox*1-*trn*L_2_ merged into *nad*2-*trn*I-*cox*1-*trn*L_2_ (Figure 4). These data suggest that mt minichromosome merging is common in *H. tuberculatus*. 

### 2.5. Phylogenetic Relationships

In the present study, the monophyly of *Haematopinus* (Haematopinidae), *Polyplax* (Polyplacidae) and *Hoplopleura* (Hoplopleuridae) was strongly supported by Bayesian inference (BI) analysis (Bpp = 0.9) and maximum likelihood (ML) analysis (Bv = 100) (Figure 5). The family Haematopinidae was sister to a clade of the families Polyplacidae + Hoplopleuridae to the exclusion of the families Pediculidae, Pthiridae and Pedicinidae with strong BI support (Bpp = 1.0) and moderate ML support (Bv = 49) (Figure 5). These results are consistent with those observed in the previous studies using nuclear genomic sequences [28,29]. In employing mt genomic datasets, however, several studies have indicated that the family Haematopinidae and families Pediculidae + Pthiridae + Pedicinidae were more closely related than to the families Polyplacidae and Hoplopleuridae with strong BI support (Bpp = 1.0), but weak support in ML analyses [19,21,30]. The mt genome is a valuable genetic marker for phylogenetic and evolutionary studies of different organisms because of its lacking of recombination, low mutation rate, and matrilineal inheritance [31,32,33]. However, recombination is found frequently in the fragmented mt genomes of Anoplura lice [27]. Recombination in mt genomes has substantial effects on phylogenetic and evolutionary studies that utilize mt genes [34,35,36]. The traditional methods for phylogenetic analysis are based on the assumption that mtDNA does not recombine; ignoring the occurrence of recombination can lead to incorrect phylogenetic reconstruction and positive selection analyses [37,38]. Collectively, our data along with others suggest that the deeper relationships among families within the parvorder Anoplura are challenge to resolve due to occurrence of recombination. Consequently, inclusion of data other than mt genomes would be greatly helpful in order to resolve phylogeny of sucking lice. 

## 3. Materials and Methods

### 3.1. Sample Collection and DNA Extraction

Adult lice *H. tuberculatus* were collected from naturally infected buffalo *Bubalus bubalis* in Khyber Pakhtunkhwa province, Pakistan. They were identified to species morphologically [1], and stored in 100% (v/v) ethanol at −40 °C after five washes in physiological saline. Total genomic DNA was extracted from ten individual lice (five females and five males) using the DNeasy Tissue Kit (Promega, Madison, USA) according to the manufacturer’s protocol. The molecular identity of each sucking louse as *H. tuberculatus* was further verified by PCR-based sequencing of regions in the mt *cox*1 and *rrn*S genes as previously described [20]. The *cox*1 gene sequences of *H. tuberculatus* were 100% identical to that of *H. tuberculatus* (GenBank accession no: EU375757) from *Bubalus bubalis* in the United Kingdom.

### 3.2. Sequencing and Assembling

DNA concentration of each sample was determined using the Qubit system (Thermo Fisher Scientific, Waltham, MA, USA). Total DNA sequencing was performed by Novogene Bioinformatics Technology Co., Ltd. (Tianjing, China) using the Illumina HiSeq2500 platform (Illumina, San Diego, CA, USA) to produce 2 × 250 bp paired-end reads and raw data were recorded in FASTQ format. The raw reads were filtered to remove containing adaptor sequences and low-quality reads (the ‘N’ percent of one end > 5%, average quality score Q < 20 and length < 75 bp after trimming) using Trimmomatic v.0.32 [39]. The mt *cox*1 and *rrn*S sequences of *H. tuberculatus* were used as the initial references to *de novo* assembled the clean reads using Geneious Prime 2020 (www.geneious.com, accessed on 1 November 2021). The assembly parameters were: minimum overlap identity 99%, maximum 3% gaps per read, maximum gap 5 bp and minimum overlap 150 bp. A circular minichromosome was identified if both ends of a contig overlapped. Previous studies [18,19] showed that the NCR are highly conserved among the mt minichromosomes of a sucking louse. The conserved NCR sequences were identified between the mt *cox*1 and *rrn*S minichromosomes and were used as references to align the clean read sequence dataset. This allowed us to extract sequence reads derived from the two ends of the coding regions of all other mt minichromosomes. We then assembled all minichromosomes individually in full length using the same method stated above for mt *cox*1 and *rrn*S minichromosome assembly.

### 3.3. Verification of mt Minichromosomes

The size and circular organization of each mt minichromosome of *H. tuberculatus* were verified by long PCR using specific primers (Appendix A), which were designed from the coding region of each minichromosome using the Primer Premier 5.0 (Premier Biosoft Interpairs, Palo Alto, CA, USA). The forward primer and reverse primer of each pair were next to each other with a small gap or no gap in between. PCR with these primers amplified each circular minichromosome in full or near full size if it had a circular organization (Appendix A). These positive amplicons were also sequenced with Illumina HiSeq2500 platform as described above. To obtain full-length and accurate sequences of the NCR of the all minichromosomes, we have re-assembled the NCR of each mt minichromosome using these obtained sequences according to the same method.

### 3.4. Annotation and Visualization

Genes were predicted with MITOS web server (http://mitos.bioinf.uni-leipzig.de/index.py, accessed on 5 November 2021) [40] and manually curated. Sequences of each protein-coding gene were then aligned against the corresponding gene of *H. suis* [13] and *H. asini* [16] using the MAFFT 7.263 software [41] to further identify gene boundaries. The location of protein-coding genes was further confirmed in ORFfinder (https://www.ncbi.nlm.nih.gov/orffinder/, accessed on 5 November 2021). Amino acid sequences of each protein-coding genes were inferred using MEGA 11 [42], and deduced amino acid sequences were used in BLAST searches of the protein database of GenBank. tRNA genes were identified using the program tRNAscan-SE [43] and ARWEN [44], and rRNA genes were identified with BLAST searches of the NCBI database and in comparison with alignments from *H. suis* [13] and *H. asini* [16]. The circular map of *H. tuberculatus* mt genome was illustrated using Microsoft PowerPoint v.2021.

### 3.5. Phylogenetic Analysis

Amino acid sequences inferred from the nucleotide sequences of 11 mt protein-coding genes common (*nad*2 and *nad*5 excluded because these genes are unidentified in *H. kitti* and *H. elephantis*) for all sucking lice (Table 3), using the elephant louse species, *H. elephantis* (GenBank: KF933032-41) as an outgroup [45]. The deduced amino acid sequences were aligned individually using MAFFT 7.122 and concatenated to form a single dataset; ambiguously aligned regions were excluded using Gblocks 0.91b using default parameters [46]. 

Phylogenetic analyses were conducted using two methods: BI and ML. BI was carried out using MrBayes 3.2.6 [47]. The most suitable model (MtArt) of evolution was selected by ProtTest 3.4 [48] at the default setting based on the Akaike information criterion (AIC). As MtArt model is a very recent addition to the models commonly used, we could not implement it in the current version of MrBayes, which used the best scoring alternative model MtREV. Four independent Markov chains (three heated and one cold) were run simultaneously for 1,000,000 metropolis coupled MCMC generations, sampling a tree every 100 generations. The first 2500 trees represented burn-in, and the remaining trees were tested for stability of likelihood values and used to compute Bayesian posterior probabilities (Bpp). We assumed that stationarity had been reached when the estimated sample size (ESS) was greater than 100, the potential scale reduction factor (PSRF) approached 1.0 and the average standard deviation of split frequencies (ASDSF) was < 0.01. ML was conducted with IQ-TREE v.2.1.3 [49]. The “Auto” option was set under the best evolutionary models, and the ML trees were constructed using an ultrafast bootstrap approximation approach with 10,000 replicates. The Bootstrap value (Bv) was calculated using 100 bootstrap replicates. Phylogenetic trees were drawn using FigTree v.1.42. 

## 4. Conclusions

The newly-described mt genome of *H. tuberculatus* presented here has a novel mt genome fragmentation pattern, differing from other three *Haematopinus* lice, proved our hypothesis. Our findings indicate that recombination plays a major role in generating the variation in the composition of mt minichromosomes among *Haematopinus* lice. Compared to other studies, our phylogeny generated from mt genome datasets showed a different topology. Therefore, inclusion of data other than mt genomes would be required to resolve phylogeny of sucking lice. This is the first report of a mt genome with ten mt minichromosomes in sucking lice, which opened new outlook into unexplored fragmentation pattern in their mt genomes. Our results would encourage further investigation on mt genome fragmentation pattern in parasitic lice and other insects.

## Figures and Tables

**Figure 1 ijms-23-13092-f001:**
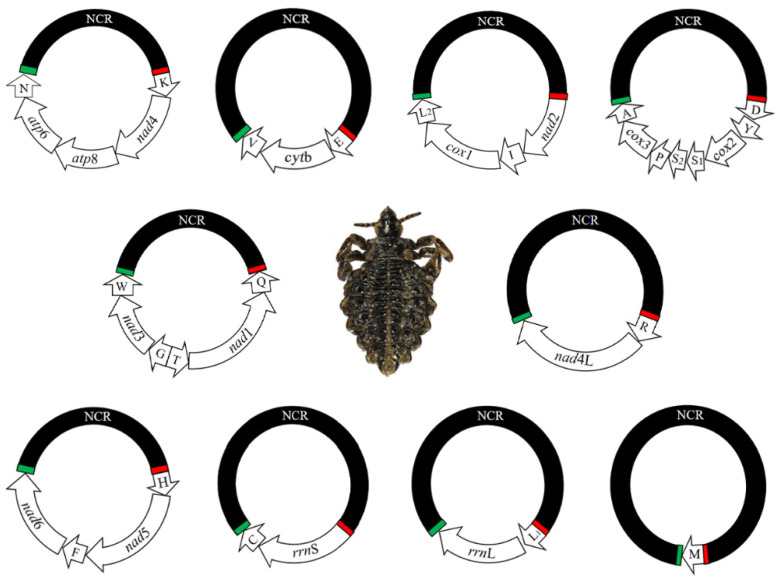
The complete mitochondrial genome of the buffalo louse, *Haematopinus tuberculatus*. Each minichromosome has a coding region and a non-coding region (NCR, in black). The names and transcript orientation of genes are indicated in the coding region and the minichromosomes are in alphabetical order of protein-coding genes and rRNA genes. Abbreviations: *atp*6 and *atp*8, ATP synthase F0 subunits 6 and 8; *co*b, cytochrome b; *cox*1–3, cytochrome *c* oxidase subunits 1–3; *nad*1–6 and *nad*4L, NADH dehydrogenase subunits 1–6 and 4L; *rrn*S and *rrn*L, small and large subunits of ribosomal RNA. tRNA genes are indicated with their single-letter abbreviations of the corresponding amino acids.

**Figure 2 ijms-23-13092-f002:**
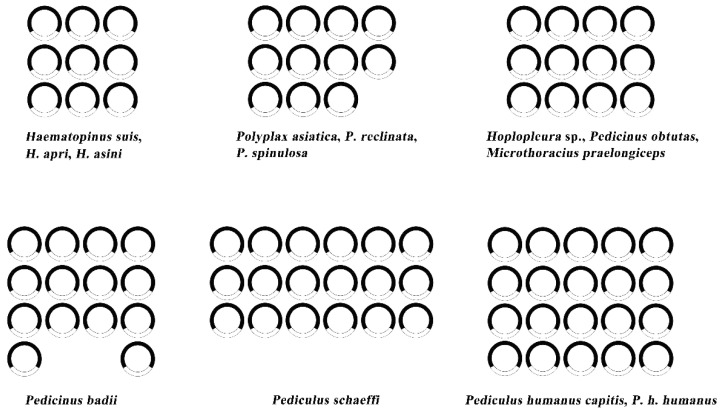
Numbers of mitochondrial minichromosomes of 12 sucking lice which all genes were identified in the mt genome.

**Figure 3 ijms-23-13092-f003:**
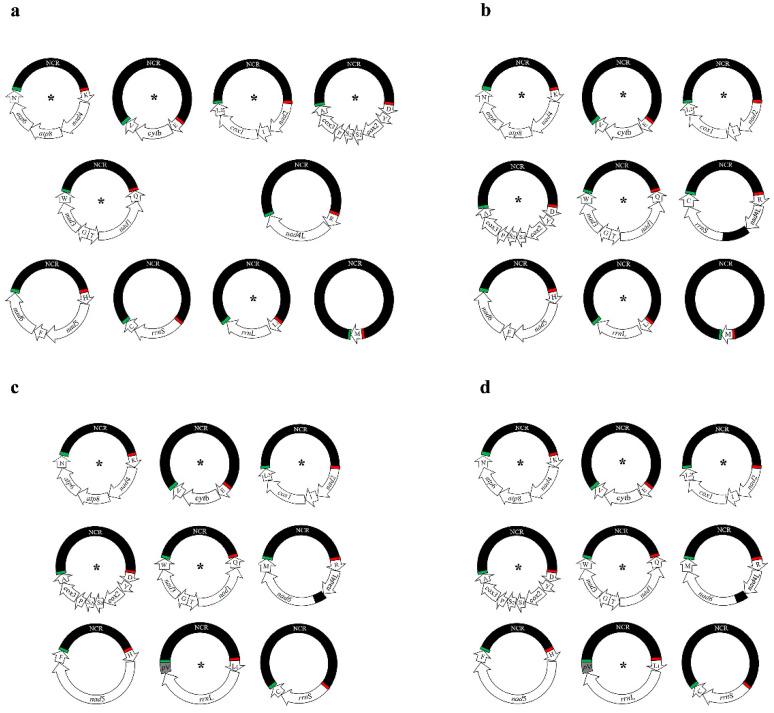
The differences among all minichromosomes of four *Haematopinus* lice. (**a**) Ten circular minichromosomes of *Haematopinus tuberculatus*; (**b**) Nine circular minichromosomes of *H. asini*; (**c**) nine circular minichromosomes of *H. suis*. (**d**) Nine circular minichromosomes of *H. apri*. See Figure 1 legend for gene name abbreviation, *p*V indicates pseudo *trn*V. * indicates the identical minichromosomes identified among four *Haematopinus* lice.

**Figure 4 ijms-23-13092-f004:**
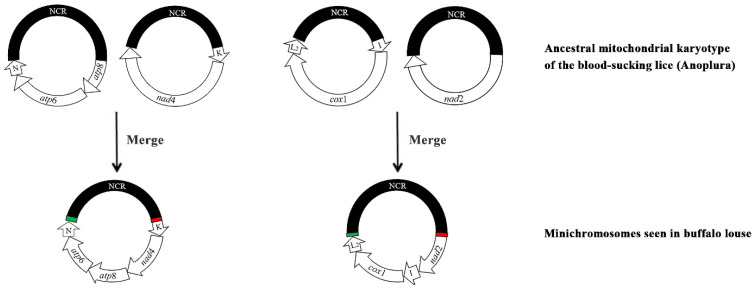
The ancestral mitochondrial minichromosomes of sucking lice that merges in *Haematopinus tuberculatus*. Gene name and transcription orientation are indicated in the coding region; non-coding regions (NCR) are in black. See Figure 1 legend for gene name abbreviation. Ancestral mt minichromosomes are inferred by Shao et al., 2017.

**Figure 5 ijms-23-13092-f005:**
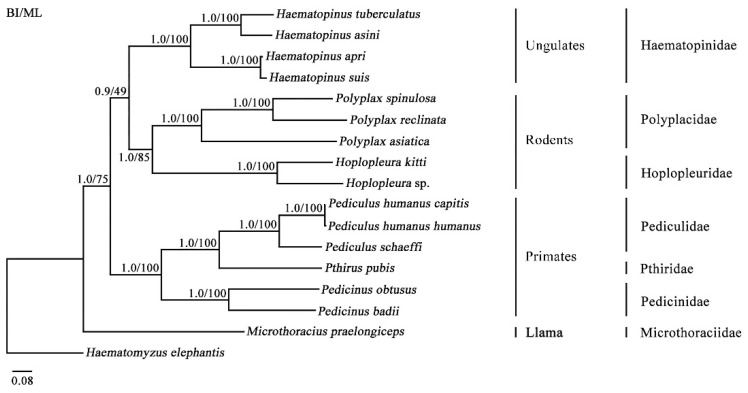
Phylogenetic relationships among 17 species of the parvorder Anopluran lice inferred by Bayesian inference method (BI) and maximum likelihood (ML) of deduced amino acid sequences of eight mitochondrial proteins using MrBayes and IQ-Tree. The elephant louse, *Haematomyzus elephantis*, was used as the outgroup. Posterior probability values (Bpp) and bootstrap values (Bv) are indicated at nodes.

**Table 1 ijms-23-13092-t001:** Mitochondrial minichromosomes of the buffalo louse *Haematopinus tuberculatus*, determined by next-generation sequencing using Illumina.

Minichromosome	Size (bp)	Size of Coding Region (bp)	Size of Non-Coding Region (bp)
*trn*K-*nad*4-*atp*8-*atp*6-*trn*N	4673	2281	2392
*trn*E-*cyt*b-*trn*V	4013	1216	2797
*nad*2-*trn*I-*cox*1-*trn*L_2_	5019	2627	2392
*trn*D-*trn*Y-*cox*2-*trn*S_1_-*trn*S_2_-*trn*P-*cox*3-*trn*A	3400	1882	1518
*trn*Q (−) -*nad*1 (−) -*trn*T (−) -*trn*G-*nad*3-*trn*W	4079	1506	2573
*trn*R-*nad*4L	4369	344	4025
*trn*H-*nad*5-*trn*F-*nad*6	4508	2,228	2280
*rrn*S-*trn*C	3586	793	2793
*trn*L_1_-*rrn*L	3882	1211	2671
*trn*M	2966	67	2899
Total	40,495	14,155	26,340

Note: minus (−) indicates the mt genes have the opposite orientation of transcription relative to the non-coding region.

**Table 2 ijms-23-13092-t002:** Long stretches of identical sequence shared between mitochondrial genes in the buffalo louse, *Haematopinus tuberculatus*.

		Long Stretches of Identical Sequence Shared (bp)																
Pairs of Genes		Buffalo Louse	Horse Louse	Pig Lice		Primate Lice						Guanaco Louse	Rodent Lice					
		Hat	Haas	Has	Haap	Phc	Phh	Ptp	Pes	Peb	Peo	Mip	Hok	Hoa	Hosp	Poa	Pos	Por
cox1	nad4	**19**	11	**12, 18**	11	**13, 18**	**13, 18**	11	14	11	11	10	14	10	13	**18**	10	11
nad4	cytb	**17**	**20**	10	10	11	11	**15**	12	14	13	11	11	10	9	11	12	11
atp8	atp6	**32**	9	9	9	10	10	8	11	11	9	11	9	10	9	8	8	9
nad4L	trnV	**14**	8	8	8	7	7	6	7	7	10	6	8	6	6	7	6	7
rrnL	rrnS	11	**10, 33**	10	9	11	11	10	12	10	11	10	9	13	9	10	10	10
trnL_1_	trnL_2_	**7, 11, 25**	**15**	**9, 10, 16**	**9, 10, 16**	**32, 33**	**32, 33**	**32, 35**	**32, 34**	**8, 14, 32**	**32, 32**	**7, 10, 27**	7	N/A	8	**28 6,**	**11, 25**	**10, 25**
trnT	trnP	7	**27**	**26**	**26**	7	7	6	8	7	7	6	7	7	6	6	7	8

Note: Abbreviations of species names are: Hat, *Haematopinus tuberculatus* (buffalo louse); Haas, *Haematopinus asini* (horse louse); Has, *Haematopinus suis* (domestic pig louse); Haap, *Haematopinus apri* (wild pig louse); Phc, *Pediculus humanus capitis* (human head louse); Phh, *Pediculus humanus humanus* (human body louse); Ptp, *Pthirus pubis* (human pubic louse); Pes, *Pediculus schaeffi* (chimpanzee louse); Peb, *Pedicinus badii* (monkey louse); Peo, *Pedicinus obtutas* (monkey louse); Mip, *Microthoracius praelongiceps* (guanaco louse); Hok, *Hoplopleura kitti* (rat louse); Hoa, *Hoplopleura akanezumi* (mouse louse); Hosp, *Hoplopleura* sp. (rat louse); Poa, *Polyplax asiatica* (rat louse); Pos, *Polyplax spinulosa* (rat louse); Por, *Polyplax reclinata* (shrew louse); N/A, not available. Stretches of shared identical sequences longer than expected by chance are in bold.

**Table 3 ijms-23-13092-t003:** The sucking lice included in phylogenetic analyses in this study.

Species	Host	GenBank Accession Number	Reference
*Haematopinus apri*	Wild pig	KC814611-19	[13]
*Haematopinus asini*	Horse	KF939318, KF939322, KF939324, KF939326, KJ434034-38	[16]
*Haematopinus suis*	Domestic pig	KC814602-10	[13]
*Hoplopleura kitti*	Rat	KJ648933-43	[14]
*Hoplopleura* sp.	Rat	MT792483-94	[20]
*Microthoracius praelongiceps*	Guanaco	KX090378-KX090389	[18]
*Pediculus humanus capitis*	Human	JX080388-407	[12]
*Pediculus humanus humanus*	Human	FJ499473-90	[11]
*Pediculus schaeffi*	Chimpanzee	KC241882-97, KR706168-69	[17]
*Pedicinus badii*	Monkey	MT721726-37	[19]
*Pedicinus obtutas*	Monkey	MT792495–506	[20]
*Pthirus pubis*	Human	JQ976018, MT721740, HM241895-8, EU219987-95	[12] [19]
*Polyplax asiatica*	Rat	KF647751-61	[15]
*Polyplax reclinata*	Shrew	MW291451-61	[21]
*Polyplax spinulosa*	Rat	KF647762-72	[15]
*Haematopinus tuberculatus*	Buffalo	OP574152-61	Present study

## Data Availability

The fragmented mitochondrial genome sequences of *H. tuberculatus* from buffalo have been deposited in the GenBank database under the accession numbers ON416547–56.

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
