# Peer review of "A Novel Mitochondrial Genome Fragmentation Pattern in the Buffalo Louse Haematopinus tuberculatus (Psocodea: Haematopinidae)"

_ijms, 2022, doi:10.3390/ijms232113092_

Round 1

Reviewer 1 Report

I am not sure if the results are valid. The authors neither shared nor intend to share the obtained raw sequencing data. Therefore all their conclusions may be invalid, I can't tell. In order to properly evaluate their work one have to see the raw data. 

Author Response

Responses to the comments and suggestions of Reviewer #1:

General comments:

I am not sure if the results are valid. The authors neither shared nor intend to share the obtained raw sequencing data. Therefore, all their conclusions may be invalid, I can't tell. In order to properly evaluate their work one have to see the raw data.

Responses: We thank the Reviewer #1’s constructive comments and suggestions on our MS. Our raw sequencing data has been deposited in the NCBI database under the accession number PRJNA883441.

Reviewer 2 Report

Fu et al present an expansion of our knowledge of mt genome fragmentation in anopluran lice.  It is a worthwhile study and mostly well conducted.  The discussion of the phylogenetic results is, however, unsupported by the results presented and comparison with other literature.  There is no reason presented to doubt the results reported (as is done in the discussion) and the cited potential reasons for phylogenetic inaccuracy are not limited to mt genomes and/or aren't relevant in this system.  Rework this section.

The manuscript overall required substantial editorial correction which I provided in an attached mark up of the manuscript.  

Author Response

Responses to the comments and suggestions of Reviewer #2:

General comments:

Fu et al present an expansion of our knowledge of mt genome fragmentation in anopluran lice.  It is a worthwhile study and mostly well conducted. The discussion of the phylogenetic results is, however, unsupported by the results presented and comparison with other literature.  There is no reason presented to doubt the results reported (as is done in the discussion) and the cited potential reasons for phylogenetic inaccuracy are not limited to mt genomes and/or aren't relevant in this system. Rework this section.

Responses: We thank Reviewer #2 very much for his/her comments on our MS. We have reworked this section. We hope you will now find the section of phylogenetic analyses clearly presented.

The manuscript overall required substantial editorial correction which I provided in an attached mark up of the manuscript.

Responses: We thank Reviewer #2 very much for his/her comments on our MS. Agreed and addressed.

Reviewer 3 Report

There are many issues with this manuscript:

1) the references are misnumbered (each line is numbered even when it belongs to one reference), which makes them impossible to track

2) the manuscript contains some sentences literally copied and pasted from other sources, which are later cited but under different numbers

3) the manuscript does not present a substantial amount of novel knowledge, the authors only sequenced one more species of a genus that is already well represented in the literature; there are some differences compared to other Haematopinus mt genomes but this is not anything ground-breaking in the whole lice-genomic scale

4) the presentation is rather poor, here are a few examples:

 4a) Lines 50-57 - the authors cumbersomely describe differences between different Haematopinus mt genomes, it would be much clearer to show their comparison in a figure (like e.g. that in Song et al. 2014)

 4b) Figure 2 - the authors just copy & paste one chromosome symbol across all the figures, it would be much better to show their relative sizes, if not separate genes. From the figure, it is not clear if the gene order is the same in all members of the group (it is not - that should be commented on)

 4c) changing of past and present tense is disturbing (e.g. lines 106-107)

 4d) Figure 3 - it is not clear where the ancestral genomes come from

 4e) There were 10 animals sequenced, the authors should show the alignments e.g. in Jiang et al., 2013, or Song et al., 2014, at least in the supplement

4f) Section 3.2 - there is a lot of information missing in this section, e.g. how did the authors check the read quality? did they trim the reads or use them raw? did they use some subsampling?

4g) Section 3.3 does not sound really clear, a figure would help

4h) Conclusion - there is no proper conclusion at all, the authors state what everyone sees. There are pieces of what could be a conclusion across the manuscript (e.g. in Introduction, lines 50-57), these should be put coherently together to make some point, which is now missing

Due to all these problems, I do not suppose that the manuscript should be published as it is. I recommend adding either a larger dataset or a deeper analysis of the current one (e.g. genes structure and location) and reworking it to a clearer form.

Author Response

Responses to the comments and suggestions of Reviewer #3:

General comments:

There are many issues with this manuscript.

Responses: We thank Reviewer #3 very much for his/her comments on our MS. We have carefully revised our article according to your excellent comments.

Specific comments:

Point 1: the references are misnumbered (each line is numbered even when it belongs to one reference), which makes them impossible to track.

Responses: Thank you for comments. Agreed and addressed. We have reworked the references to ensure one reference belongs to one number.

Point 2: the manuscript contains some sentences literally copied and pasted from other sources, which are later cited but under different numbers.

Responses: Thank you for comments. Agreed and addressed. We have carefully revised these sentences according to your excellent comments.

Point 3: the manuscript does not present a substantial amount of novel knowledge, the authors only sequenced one more species of a genus that is already well represented in the literature; there are some differences compared to other Haematopinus mt genomes but this is not anything ground-breaking in the whole lice-genomic scale.

Responses: We thank Reviewer #3 very much for his/her comments on our MS. We acknowledge that mitochondrial (mt) genomes of three species (pig louse, wild pig louse and horse louse) have been sequenced from the genus Haematopinus. However, the buffalo louse H. tuberculatus cleanly showed a novel mitochondrial genome fragmentation pattern. We also acknowledge that papers describing fragmented mt genomes have similarities in structure, but this is the nature of such investigations. Nonetheless, the results described here are original and pertain to an important ectoparasite that has received extremely limited attention, and deserves more. In the present manuscript, we go beyond the description of one fragmented mt genome and we also undertake the recombination of mt minichromosomes in the Haematopinus lice and phylogenetic analysis of sucking lice. In our opinion, the MS provides new and important insights into a fragmented mt genome, which opens new outloo into unexplored mt genome fragmentation pattern in sucking lice as well as future phylogenetic investigations of broad significance. Therefore, this study underpins critical, future genomic studies, such that we consider it to be significant and publishable in International Journal of Molecular Sciences.

Point 4: Lines 50-57 - the authors cumbersomely describe differences between different Haematopinus mt genomes, it would be much clearer to show their comparison in a figure (like e.g. that in Song et al. 2014).

Responses: We thank Reviewer #3 very much for his/her comments on our MS. We have added a new figure to show differences between different Haematopinus mt genomes. See figure 3. Many thanks again.

Point 5: Figure 2 - the authors just copy & paste one chromosome symbol across all the figures, it would be much better to show their relative sizes, if not separate genes. From the figure, it is not clear if the gene order is the same in all members of the group (it is not - that should be commented on).

Responses: We thank Reviewer #3 very much for his/her comments on our MS. In our opinion, Figure 2 show different numbers of minichromosomes among sucking lice. In addition, considering that this reviewer wanted to add their relative sizes and gene order of minichromosome, describing their relative sizes and gene order of minichromosome would not add value to the manuscript because such information has been showed in previous literatures (Jiang et al., 2013, Song et al., 2014).

Point 6: changing of past and present tense is disturbing (e.g. lines 106-107).

Responses: Thank you for comments. Agreed and addressed.  

Point 7: Figure 3 - it is not clear where the ancestral genomes come from.

Responses: We thank Reviewer #3 very much for his/her comments on our MS. We have cited the reference in figure legend.

Point 8: There were 10 animals sequenced, the authors should show the alignments e.g. in Jiang et al., 2013, or Song et al., 2014, at least in the supplement.

Responses: We thank Reviewer #3 very much for his/her comments on our MS. In Jiang et al., 2013, Song et al., 2014, they have only aligned of the consensus sequences of the full-length NCRs of the mt minichromosomes. In our opinion, alignment of the mitochondrial genome sequences of 10 sequenced sucking lice from animals would not add value to the manuscript.

Point 9: Section 3.2 - there is a lot of information missing in this section, e.g. how did the authors check the read quality? did they trim the reads or use them raw? did they use some subsampling?

Responses: We thank Reviewer #3 very much for his/her comments on our MS. Agreed and addressed. We have described detailed the information in this section.

Point 10: Section 3.3 does not sound really clear, a figure would help.

Responses: We thank Reviewer #3 very much for his/her comments on our MS. We have added a new figure as figure S1.

Point 11: Conclusion - there is no proper conclusion at all, the authors state what everyone sees. There are pieces of what could be a conclusion across the manuscript (e.g. in Introduction, lines 50-57), these should be put coherently together to make some point, which is now missing.

Responses: We thank Reviewer #3 very much for his/her comments on our MS. Agreed and addressed. According to our findings, we have revised the conclusions.

Point 12: Due to all these problems, I do not suppose that the manuscript should be published as it is. I recommend adding either a larger dataset or a deeper analysis of the current one (e.g. genes structure and location) and reworking it to a clearer form.

Responses: We thank Reviewer #3 very much for his/her comments on our MS. We have carefully revised our MS according to your excellent comments. We have addressed individual points to the best of our knowledge and ability. In addition, we also have done an in-depth analysis of novel mitochondrial fragmentation pattern and compared to other Haematopinus mt genomes. We consider that the present MS has been significantly enhanced.

Round 2

Reviewer 3 Report

I appreciate that the authors dedicated substantial effort to improving the manuscript. In the present form, there are still minor issues detectable, e.g. the conclusions are still rather short and unsound, and the authors addressed some of my comments only partially (e.g., they are still hiding which software they used for the filtering). But that is of minor importance provided other reviewers will not find more significant errors. The presentation of the results is much clearer now, so it can be published as it is.

Author Response

Responses to the comments and suggestions of Reviewer #3:

General comments:

I appreciate that the authors dedicated substantial effort to improving the manuscript. In the present form, there are still minor issues detectable, e.g. the conclusions are still rather short and unsound, and the authors addressed some of my comments only partially (e.g., they are still hiding which software they used for the filtering). But that is of minor importance provided other reviewers will not find more significant errors. The presentation of the results is much clearer now, so it can be published as it is.

Responses: We thank the Reviewer #3’s constructive comments and suggestions on our MS. We have added more information into conclusions to ensure that it is sound. We have added software which used for the filtering. In addition, we have also gone through the MS to make sure the overall information is clear and that there is proper grammar and word usage. We hope you will now find the MS clearly presented.